# [Re] Satellite Image Time Series Classification with Pixel-Set Encoders and Temporal Self-Attention

## Reproducibility Summary

*The presented study evaluates "Satellite Image Time Series Classification with Pixel-Set Encoders and Temporal Self-Attention" by Garnot et al. (2020) within the scope of the ML Reproducibility Challenge 2020. Our work focuses on both aspects constituting the paper: the method itself and the validity of the stated results. We show that, despite some unforeseen design choices, the investigated method is coherent in itself and performs the expected way.*

**Scope of Reproducibility**

The evaluated paper presents a method to classify crop types from multispectral satellite image time series with a newly developed *pixel-set encoder* and an adaption of the Transformer (Vaswani *et al.*, 2017), called *temporal attention encoder*.

**Methodology**

In order to assess both the architecture and the performance of the approach, we first attempted to implement the method from scratch, followed by a study of the authors' openly provided code. Additionally, we also compiled an alternative dataset similar to the one presented in the paper and evaluated the methodology on it.

**Results**

During the study, we were not able to reproduce the method due to a conceptual misinterpretation of ours regarding the authors' adaption of the Transformer (Vaswani *et al.*, 2017). However, the publicly available implementation helped us answering our questions and proved its validity during our experiments on different datasets. Additionally, we compared the papers' temporal attention encoder to our adaption of it, which we came across while we were trying to reimplement and grasp the authors' ideas.

**What was easy**

Running the provided code and obtaining the presented dataset turned out to be easily possible. Even adapting the method to our own ideas did not cause issues, due to a well documented and clear implementation.

**What was difficult**

Reimplementing the approach from scratch turned out to be harder than expected, especially because we had a certain type of architecture in mind that did not fit the dimensions of the layers mentioned in the paper. Furthermore, knowing how the dataset was exactly assembled would have been beneficial for us, as we tried to retrace these steps, and therefore would have made the results on our dataset easier to compare to the ones from the paper.

**Communication with original authors**

While working on the challenge, we stood in E-mail contact with the first and second author, had two online meetings and got feedback to our implementation on GITHUB. Additionally, one of the authors of the Transformer paper (Vaswani *et al.*, 2017) provided us with further answers regarding their models' architecture.

# 1 Introduction

The *machine learning* community showcases impressively and with great success how to design systems for analysing large data inventories, with the goal of identifying relevant patterns within them. At the same time, the *remote sensing* and *Earth observation* sector found itself confronted with the availability of novel dedicated sensor platforms. These are now capable of continuously acquiring new data at high temporal, spatial, and spectral frequencies, requiring the development of innovative and efficient ways to process these data stocks. In light of that, several machine learning methods have found wide application in the field of remote sensing and Earth observation. As the availability of observation data increased massively during the last decades (Lary *et al.*, 2018), various retrieval, detection, and prediction problems can be addressed this way. Nevertheless, Earth observation data is mostly of a very inhomogeneous nature, which is due to the different designs and layouts of the receiving sensor platforms. In addition, geophysical processes on the Earth's surface are complex and manifest themselves in observable changes with different dynamic patterns. Therefore, the goal of gaining deeper understanding of such processes requires the use of interpretable models (Maxwell, Warner, and Fang, 2018).

In their recent CVPR publication "Satellite Image Time Series Classification with Pixel-Set Encoders and Temporal Self-Attention", Garnot *et al.* (2020) propose a new method to address these issues. Motivated by the practical problem of *crop type classification* from sequences of optical satellite imagery—that we consider a proxy for the entirety of vegetative processes on the Earth' surface—, the authors made use of *attention mechanisms*. In particular, they claimed adapting the *Transformer* architecture (Vaswani *et al.*, 2017) that has gained considerable popularity in the recent past, enabling it to digest such specific Earth observation data modalities. Additionally, they introduce a *pixel-set encoder* as a new option to deal with medium-resolution satellite images instead of the known convolutional neural networks in image processing.

Due to their aforementioned properties, the handling of Earth observation data in practical applications requires special care. Most prevalently, they exhibit a considerable amount of spatial autocorrelation (Spiker and Warner, 2007), reinforcing the already known issues of *underspecification* inherent to data-driven machine learning models (D'Amour *et al.*, 2020). This generally leads to overfitting effects and poor generalisation performance. Hence, we carefully reproduced the proposed methods, first by starting from scratch just following the descriptions given in the paper under investigation, and subsequently by adapting the reference implementation openly provided by the original authors. To sanity-check both implementations and to further assess the transferability and generalisation properties of the studied model, we carried out further experiments relying on an alternative but comparable dataset. Following the idea of this reproducibility challenge, we will make our implementation and data public, allowing the community to likewise evaluate our findings.

## 1.1 Reproducibility questions

As a foundation of our reproduction study, we identified the following key questions, each one examining one particular aspect or claim of the original paper:

 *i)* Is it possible to reproduce the presented methods and their performance with *and* without referring to the authors' publicly available code?

 *ii)* To which extent do the author's implicit claim of adapting the transformer architecture affect the model and its performance?

 *iii)* Does the model perform comparably well when being only tested or both trained and tested on a different dataset?

 *iv)* How is the outcome influenced by the choice of the test set assembly, namely by splitting the data randomly or regionally?

## 1.2 Contributions beyond the original paper

In addition to the results in the original paper, we also report the performance of the method evaluated on an alternative dataset and on a test set that does not show regionally overlap with the training set.

# 2 Methodology and experimental setup of the reproduction study

In order to study the reproducibility of the original publication, we followed different approaches to answer the questions raised in Section 1.1. Each of the following subsections will address one of these questions by introducing

the methodology behind the chosen experiments and present our obtained results as replies to the author's claims. Generally, we traversed the following three different experimental stages that we will also publish on GITHUB[1] for better traceability: We first started by developing the entire approach ourselves in PYTHON and PYTORCH (Paszke *et al.*, 2019), as described in Section 2.1.1, followed by a short study on the original implementation in Section 2.1.2 and a comparison of one particular technical aspect in Section 2.2. Eventually, we conducted experiments on the influence of input data in the remaining sections.

## 2.1 Reproduction and accuracy of the satellite time series classification

The proposed architecture for *satellite time series classification* consists of two components which the authors introduce as the *pixel set encoder (PSE)* and the *temporal attention encoder (TAE)*. While the former takes care of a randomly sampled pixel set from a crop parcel and produces an embedding of the input, the latter, an adapted variant of the Transformer architecture (Vaswani *et al.*, 2017), produces an output by applying self-attention to these multi-temporal embeddings. Unlike practices familiar from *natural language processing*, PSE and TAE get both optimised during the training phase. A detailed summary of the composition and number of parameters in the networks can be obtained from *Table 1* in the studied paper.

All experiments were conducted on an UBUNTU 20.10 workstation equipped with 64 GB of RAM, an INTEL I7-8700 CPU and an NVIDIA GEFORCE RTX 2060 GPU.

### 2.1.1 Full replication study of the approach

We reimplemented the proposed architecture described in *Section 3: Methods* of the investigated paper almost literally in PYTHON. As the section is subdivided into the three parts describing the *spatial encoder*, also referred to as the *pixel-set encoder*, the temporal attention encoder, and the *spatio-temporal classifier*, these three modules likewise build the core of this reproduction study. More precisely, *Table 1* of Garnot *et al.* (2020) allowed us to inherit all model hyper-parameters straightforwardly. For one of the four used *multi-layer perceptrons (MLPs)*, it was stated that it consisted of *fully-connected layers* FC, *batch normalisation*, and *ReLU activations*. We, thus, inductively assumed these components to be part of the other three MLPs as well. While we managed to develop an inefficient yet working version of the spatial encoder, some aspects of the *temporal attention encoder* appeared unintuitive at first sight. Unlike the original Transformer model (Vaswani *et al.*, 2017), on which this module is based, the values $v$ are not calculated by a fully-connected layer. Instead, the sum of the *spatial encoder* outputs and the positional encoding is multiplied with the attention mask $a$, which is visualised in Figure 1a. The authors motivated this change with the claim that it *"removes needless computations, and avoids a potential information bottleneck"* (*cf.* Garnot *et al.*, 2020, Section 3.2.). Having the original Transformer implementation in mind, we misinterpreted this design choice. Thus, in our implementation, we divided the multi-head attention input into four equal tracks, as we were not able to think of another way to end up having 512 nodes to be passed over to the $\mathrm{MLP}_3$ (*cf. Table 1* of Garnot *et al.*, 2020). This misconception was partially reinforced by the superscript $(h)$ in formula (5) in Garnot *et al.*, 2020, *i.e.*,

$$k_h^{(t)}, q_h^{(t)} = \mathrm{FC}_1^{(h)} \left( e^{(t)} + p^{(t)} \right) \quad , \tag{1}$$

suggesting that, for each head $h$, an entirely independent fully-connected layer $\mathrm{FC}^{(h)}$ was used. This way, our network reimplementation became incredibly blown up and we were not able to spot the correct approach.

All hyper-parameters and training details were directly taken from the *Section 3.4: Implementation details* from Garnot *et al.* (2020). After completing the first presented stages of the implementation, we were able to achieve an accuracy of about 60 %. Subsequently, we had a first online meeting with the first and second author of the investigated paper, where we identified some misconception concerning the used labels: Instead of using all 20 classes from the `label_19class` dictionary in the provided `lables.json` file, only the top-20 classes from the `label_44class` dictionary of the same file were utilised by the authors, *i.e.*, the classes with more than 100 occurrences in the dataset. It also got to our attention that, *in lieu* of the proposed batch normalisation, the multi-layer perceptrons should perform *layer normalisation*. Unfortunately, despite the first author providing helpful feedback on our implementation via GITHUB, we were still not able to achieve a relevant increase in accuracy compared to the previously stated 60 %. After comparing our implementation to the reference implementation provided by the authors, it became apparent that we struggled to grasp the authors' ideas about the data organisation and the spatio-temporal classifier. Therefore, we will investigate what led us to misinterpret the Transformer's adaption in Section 2.2 and the data organisation in Section 2.3.2.

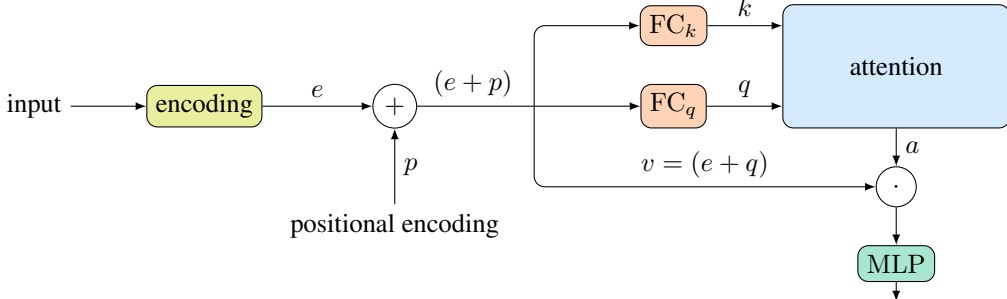

(a) The temporal attention encoder according to the presented paper, where the values $v$ are not calculated specifically, but the sum of $e$ and $p$ is directly passed through to the dot product.

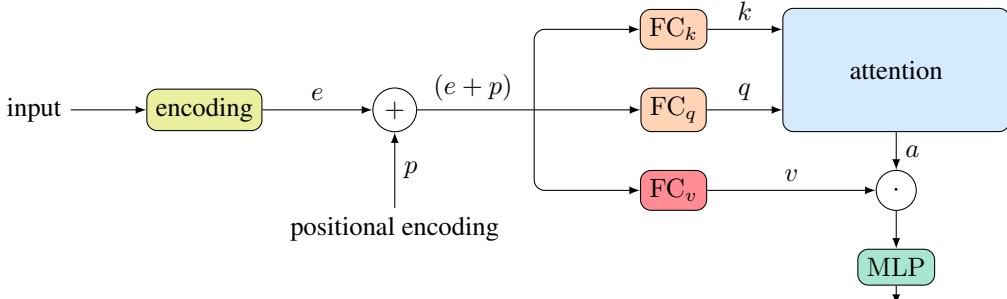

(b) Our adaption of the temporal attention encoder with the additional fully-connected layer $FC_v$, as mentioned in "Attention is All you Need" by Vaswani *et al.* (2017).

Figure 1: Illustration of the architectural difference between (a) the proposed temporal attention encoder and (b) the extended version of it presented in this study. While Garnot *et al.* approached satellite image time series classification with a lightweight adaption of the Transformer architecture (Vaswani *et al.*, 2017), we kept the original third fully-connected layer $FC_v$. The additional building blocks of the surveyed method—namely the encoder, positional encoding, attention, and final MLP—remained unchanged. This schematic representation does not include the multi-head attention, which mainly influences the *attention* module and would increase the complexity unnecessarily.

### 2.1.2 Evaluation of the original implementation

Obtaining the authors' code and running it locally proved to be easy thanks to a well-documented GITHUB repository[2]. In general, their reference implementation is modular, clearly structured, and sufficiently commented which makes the entire architecture easy to adapt to one's own needs. The model can either be trained from scratch or, together with a provided checkpoint, used to solely run the inference. We present a comparison of the test results from the checkpoint and when training from scratch in Table 1a. Using the system specified in Section 2.1, training the model for one single epoch took approximately 27 s, while running the inference on one sample batch containing 128 parcels completed within about 0.04 s. For the following experiments, we exclusively used the official reference implementation to ensure comparability.

### 2.2 Experiments on the transformer architecture

As stated in Section 2.1.1, we previously relied on some faulty assumptions related to the architecture of the adapted Transformer multi-head attention. Fortunately, the original code provided by the authors helped us to reconstruct their initial idea. Nevertheless, having the vanilla off-the-shelf Transformer implementation in mind, we were inquisitive about the impact of the particular architectural change proposed by Garnot *et al.* Hence, we took the reference implementation and changed two lines of it in a way that the temporal attention encoder then employed a third fully-connected layer $FC_v$, just like the vanilla Transformer attention model does, as described by Vaswani *et al.* (2017). Figure 1b illustrates this subtle change in comparison to the architecture realized by the authors of the paper under investigation, shown in Figure 1a. We provide the code for this version as a fork of the original repository.[3]

---

[1] https://github.com/maja601/RC2020-psetae

[2] https://github.com/VSainteuf/pytorch-psetae

[3] https://github.com/maja601/pytorch-psetae

Table 1: Covering several experiments, (a) and (b) show the results of the presented temporal attention encoder (TAE) and our adaptation of it with the additional fully-connected layer $FC_v$, respectively. We therefore hoped to give an impression of the performance of the models under different circumstances and also the impact of the architectural change we evaluated in Section 2.2.

(a) Comparison of the original TAE and its adaption with $FC_v$. The overall accuracy on FR-T31TFM for the approach of Garnot *et al.* is given twice: once it was obtained from the checkpoint provided by the authors and once by training from scratch. The last column indicates how many trainable parameters each of the network variants comprises.

| Model variant | Overall accuracy | | Number of parameters |
|---|---|---|---|
| | checkpoint | trained | |
| TAE (Garnot *et al.*, 2020) | 94.19 | **94.26** | 164 116 |
| TAE with $FC_v$ (Vaswani *et al.*, 2017) | — | 94.24 | **131 476** |

(b) Analogous to (a), both versions of the models were trained on FR-T31TFM, but this time tested on the unseen region SI034 of the SI-T33TWM dataset. This practice is often referred to as cross-dataset evaluation.

| Model variant | Overall accuracy | |
|---|---|---|
| | checkpoint | trained |
| TAE (Garnot *et al.*, 2020) | 61.75 | 61.65 |
| TAE with $FC_v$ (Vaswani *et al.*, 2017) | | **62.03** |

While this adaption of the proposed architecture reduces the number of trainable model parameters to 80 %, we were able to reproduce the reported performance or even experienced a slight increase in accuracy, as summarised in Table 1a.

## 2.3 Generalisation and transferability

Until that stage, we mainly considered the theoretical aspects of this reproduction study. Conversely, this section focuses on the application-oriented side. In light of our observations described before, we wanted to investigate whether the increased number of parameters in the original model can become beneficial when scaling the underlying classification problem by confronting it with an alternative dataset.

### 2.3.1 Datasets

Although a tremendous amount of satellite data, especially from the SENTINEL-2 platforms, is publicly available, the computer vision and machine learning community still lacks labels or annotations for addressing most relevant research questions. Therefore, Garnot *et al.* (2020) did not only publish a new method, but also complemented it with a dataset containing crop type labels of more than 190 000 agricultural parcels within the area of a particular tile of the SENTINEL-2 tiling grid T31TFM, located in France. Additionally and to evaluate the reproducibility of the presented methods' results on a different input, an analogous dataset with parcels located in Slovenia was constructed in the course of this study. The following two sections gives insight into the background and properties of these two datasets.

**FR-T31TFM: Dataset from the paper**   Unlike CNN-based methods, the approach by Garnot *et al.* does not require the observation data to be stored as images with defined neighborhood relations, but rather as an unordered set of pixels for each parcel. From their GITHUB page, a toy dataset containing 500 parcels, each saved as NUMPY data files, can be obtained. To get access to the entire dataset, an inquiry needs to be sent to the authors, which they reply to within no time. This dataset includes 192 056 NUMPY data files of dimension $T \times C \times N$, with $T$, $C = 10$, and $N$ being the number of observations dates, spectral bands, and pixels for each particular parcel, respectively. It additionally comes with several metafiles, *i.e.*, the dates of the observations, the labels of the parcels, the geometric features of the parcels—which are stated to be necessary for the pixel set encoder—, and pre-computed normalisation values. By design, the dataset is randomly partitioned into test, validation, and training parcels, following a split ratio of 3:1:1.

This dataset also faces one of the biggest challenges in crop type classification from satellite data, namely the uneven distributed data foundation, as visualised in Figure 3a.

In their original paper, Garnot *et al.* refer to several data preprocessing steps, such as reducing the number of spectral bands delivered by the SENTINEL-2 satellite from 13 to 10, linearly interpolating ground pixels that are affected by cloud cover, normalising the reflectance data, and adding Gaussian noise.

**SI-T33TWM: Additional dataset**   As part of another project at our lab, a pan-European reference dataset for crop type classification is currently under development and will be made publicly available early next year. This way, we had the chance to use some of the data obtained from Slovenia to construct a pixel set similar to the one presented by Garnot *et al.* (2020). In order to keep it as close as possible to the original, we selected similar time steps and

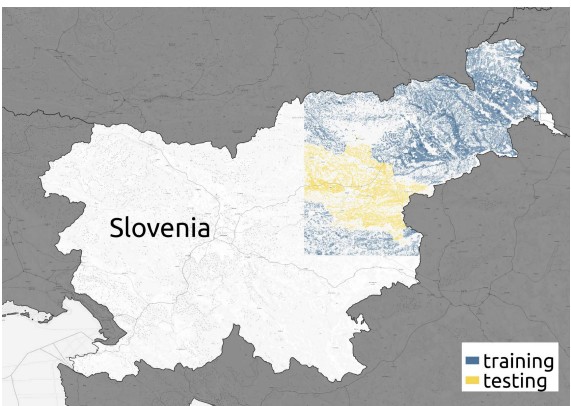

Figure 2: Schematic illustration of the additional SI-T33TWM dataset located in Slovenia with reference crop parcels mainly covering the north-eastern part of the country. While the blue areas are used to evaluate the train and test procedure proposed by Garnot *et al.*, the yellow area was set aside in advance. This way, we ensured to have an extra regionally differentiated test set that does not suffer from the issue of spatial autocorrelation.

also performed a linear cloud pixel interpolation on L2A SENTINEL-2 data. As we did not have access to the precise cloud detection module used by the authors, we manually annotated cloudy pixels in this region of interest. The `dataset_preparation.py` script provided by Garnot *et al.* took care of pooling the Sentinel-2 data and the reference data in GEOJSON format. The necessary normalisation parameters had likewise to be calculated manually, as well as the operation to extract the geometric features used in the spatial encoder. Contrasting the description given in the paper under investigation, the sequential order of components within this geometry feature vector differed in a way that its second element appeared not to be the *pixel count $N$* but rather, as we assume from looking at the original dataset, the area of the bounding box. Considering the labels, we prepared two versions: One file with the previously stated top-20 crop classes from the original dataset, called top-20-F, and another one containing the top-20 classes from our alternatively chosen region, analogously named top-20-S and illustrated in Figure 3b. Since the crop cultivation in Slovenia differs from France, several classes of the original top-20-F were not represented in the new dataset, as shown in Figure 3c. This appears to render the classification on top-20-F an easier problem than on top-20-S. Section 2.4 will provide more background regarding the split of the dataset and to one of the particular difficulties inherent to geospatial data, which is why we put one region aside in advance. This way, we were able to evaluate the performance of the method also on regionally unseen data. A visual explanation of the train and test split is shown in Figure 2.

### 2.3.2 Cross-dataset evaluation

The first experiment on the generalisation abilities of the described method was performed with the model being trained on the FR-T31TFM dataset. After the descriptions of the chosen classes were obtained from the original authors, we picked the same classes from our SI-T33TWM dataset and ran the inference on our prepared test pixel set. Table 1b summarises the overall results, showing that the performance dropped significantly compared to the ones reported previously (*cf.* Table 1a).

### 2.3.3 Method application to a different region

Taking advantage of the relatively short training time, it was easily possible to train the entire model on our own data. This way, we evaluated whether the outcomes reported by Garnot *et al.* could be reproduced, even without having access to data at exactly the same preprocessing level. For this purpose, we ran the experiment twice on the new SI-T33TWM dataset, *i.e.*, first with the top-20-F labels and then with the top-20-S labels. In Table 2, the columns *random split* show the results of these experiments. These can directly be compared to the ones from the paper.

### 2.4 Experiments on the choice of the test set

Beside the issue of limited geometric resolution in SENTINEL-2 data, the influence of spatial autocorrelation has always to be taken into account when dealing with Earth observation satellite imagery (Spiker and Warner, 2007). Due to the coarse sampling of the Earth's surface, adjacent pixels might share relevant amounts of information with each other. Therefore, using contiguous field parcels for training as well as for testing can lead to non-representative results over-estimating the true performance of the classification model. We tried to account for these effects by reserving an

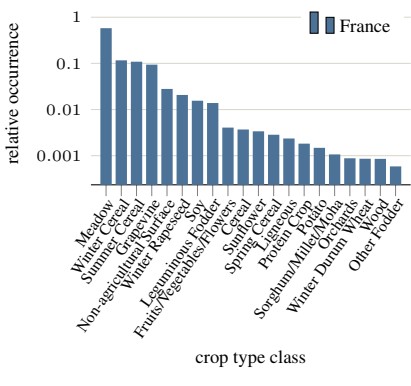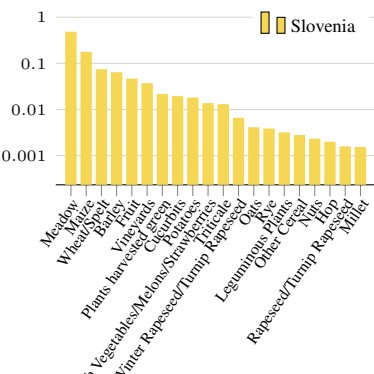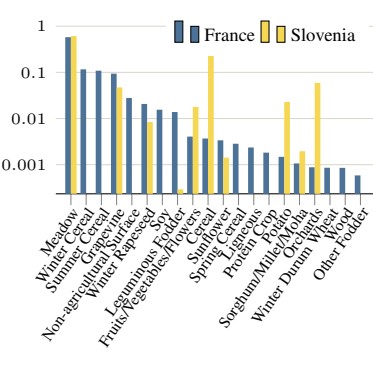

(a) The relative class occurrence of the crop types in the FR-T31TFM dataset provided by the authors of the reproduced paper.

(b) The class distribution of 20 most frequent classes of the newly constructed SI-T33TWM dataset.

(c) A comparison of the in (a) shown class distribution and the relative occurrence of these classes in the new SI-T33TWM dataset. This shows that training and testing on the SI-T33TWM while using the top-20-F labels will lead to fewer classes in total.

Figure 3: Statistics on the different datasets used in this study. As the data harmonisation was done by hand, the class names of one dataset might include differently named crop types from the other dataset and vice versa. We point out that the relative occurrence is indicated in log-scale, highlighting the strong class imbalance towards meadow.

Table 2: Results of the proposed method and our adapted version of it when being trained on SI-T33TWM. Generally, we evaluated four different scenarios where we used two different label files with each time two different ways of splitting the test set from the training set. The labels either represented the top 20 crop type classes found in the FR-T31TFM dataset (top-20-F) or the top 20 classes from the region of SI-T33TWM (top-20-S). As we set the parcels from one region of Slovenia aside, we were able to evaluate the methods not only on the proposed randomly drawn, but also on a regionally separated test set. It is necessary to recollect that with the stated top-20-F classes the entire dataset has less crop types to classify, which is illustrated in Figure 3c.

| Model variant | top-20-F classes | | top-20-S classes | |
| --- | --- | --- | --- | --- |
| | random split | regional split | random split | regional split |
| TAE (Garnot *et al.*, 2020) | 90.92 | 89.80 | 87.42 | 83.84 |
| TAE with $FC_v$ (Vaswani *et al.*, 2017) | 90.88 | 89.50 | 87.50 | 83.86 |

entirely separate region, shown in Figure 2, as an additional test set. This region was selected to approximate one-fifth of the parcels and a class distribution representative for the entire SI-T33TWM dataset. Results of this experiment are included into Table 2, where we compare the original random split to the new regional split.

## 3 Discussion

This section focuses on the analysis of reproducibility in general and will not justify the authors claim that their method is the current state-of-the-art approach to solve satellite time series classification. We therefore split the findings of the study into two aspects that we tried to evaluate: On the one hand, we will discuss the insights we have gained by reproducing the methodological process itself, and on the other hand, we will elaborate our approach to reproduce the desired results produced by the method.

### 3.1 Reproducibility of the method

When reimplementing the full architecture, we found that, despite having all parameters at hand, it would have been helpful to have access to more information concerning the data preprocessing and organisation, as well as to the adaption of the original Transformer model, to achieve performance similar to that reported by the authors. From our perspective,

we cannot tell whether or not better PYTHON programming skills would have been beneficial and if someone with more experience with the multi-head attention would have been able to understand and implement the method right away. In any case, the authors certainly developed a coherent methodology and, by providing the corresponding code alongside with the paper, have ensured that all interested parties can clearly follow their ideas.

As a result of our misinterpretation of the Transformer adaptation proposed by Garnot *et al.*, we came across an alternative approach that performed comparably well as the one from the paper, but requiring only 80 % of its parameters. When we asked the authors about this observation, we concluded that we all had different opinions on the most obvious derivative of the method developed by Vaswani *et al.* applied to the problem of satellite time series classification. Upon request, the authors of the Transformer paper confirmed that keeping the fully-connected layer $FC_v$ has proven to be helpful and acknowledged the validity of our approach. Under these circumstances, it is not straightforwardly answerable whether the implicit claim of having the Transformer adapted in the papers' way can be supported.

However, it can be said that the well-documented code and clean GitHub repository contributed strongly to our understanding of the method and helped us answer most of our comprehension question. Since this all is publicly available, a reproduction of the presented method based on that implementation is possible.

## 3.2 Reproducibility of the outcomes

Besides the possibility to reproduce the methodological aspects of the original paper, we were also interested in whether we could achieve the results stated in the investigated paper. By training the entire model by ourselves with the original dataset, we faced no considerable difficulties using the data. We were, in fact, able to achieve a slightly higher overall accuracy on the original test set compared to using the model pre-trained by the authors.

However, a slight drop in performance became observable when we used an alternative yet similarly preprocessed dataset. While we still reached an overall accuracy of over 87 % for the random split on our new and potentially more challenging dataset, we were not able to reach 84 % when running the inference on a regionally separated test set. This result highlights the importance of the right choice of a representative test set, especially when using Earth observation imagery, while still acknowledging that the presented method is potentially able to generalise to unseen and new data. Nevertheless, our reproduction experiments confirmed the claimed validity of the approach and it is to be left to the authors and the entire research community to investigate whether the presented model is able to compete with state-of-the-art methods given broader and more diverse datasets.

The only issue with the presented method arose when we used one dataset for training and another one for testing. Although we tried to preprocess our new dataset exactly the way the authors did, we were not able to obtain convincing results. There might be several reasons causing this issue, like an incorrect harmonisation of the dataset, other parameters for the interpolation of the cloud-covered pixels, or assumptions about the data we unknowingly and implicitly made different than the authors. Hence, it remains interesting for us to know how exactly the data was processed.

To summarise, we support the claim that the method can successfully classify crop parcels when it was trained on data that was acquired under the same conditions, as the data it eventually gets tested on.

# 4 Conclusion

During the proposed study of "Satellite Image Time Series Classification with Pixel-Set Encoders and Temporal Self-Attention" by Garnot *et al.* (2020), we assessed several questions regarding different aspects of the reproducibility of the paper. Therefore, we first attempted to reimplement the methodology from scratch based on the descriptions given in that paper. As this proved to be more challenging than expected and prone to misunderstandings, we proceeded to evaluate the provided clean implementation in terms of an adaption of the Transformer architecture (Vaswani *et al.*, 2017). There, we came across a discrepancy between our understanding of the vanilla multi-head attention concept and the one used in the paper. Our obtained results show that, by changing the proposed adaptation in a subtle way towards the more basic multi-head attention, the model uses considerably fewer parameters, while still performing equally well.

When employing the authors' implementation and dataset, we were able to reproduce the presented results straightforwardly and even on a new dataset that we specifically developed for this survey, the approach delivered meaningful results. The only issues arose when the training and the test dataset did not share exactly the same properties lifting the accurate preprocessing of the data to a crucial component of the proposed method.

In conclusion, we can state that the examined method is conclusive in itself and valid. Our experiments speak in favour of the approach and our findings might highlight a path in which further works should proceed. This direction of research could take advantage of the dataset which we will make publicly available within the next few months.

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
