# OpenReview forum: "[Re] Satellite Image Time Series Classification with Pixel-Set Encoders and Temporal Self-Attention"
_ML_Reproducibility_Challenge/2020 — RC2020_

### Official Review · AnonReviewer3 · 2021-02-28
**Few details missing**

**Rating:** 8
**Confidence:** 4

**Review:**

The proposed work reproduces the result from "Satellite Image Time Series Classification with Pixel-Set Encoders and Temporal Self-Attention (CVPR 2020)". The proposed work was able to reproduce the result (except for different dataset) from the original paper. However, the consideration of the following points might improve the quality of the proposed work

1) A thorough hyperparameter search and a corresponding discussion is expected.

2) The original paper also demonstrates that the processing time and memory requirement is considerably decreased. It would be better to comment/validate this statement.

**Familiar With The Original Paper:**

I have read the original paper

**Reproducibility Summary:**

Report has summary

---

### Official Review · AnonReviewer2 · 2021-03-01
**Well written and insightful reproducability report**

**Rating:** 8
**Confidence:** 4

**Review:**

This report aims to reproduce a paper on classification of time sequences of satellite imagery using transformers. The report describes the attempt to reproduce the original paper from scratch, then using the provided code. In addition, the report adds on more investigation on another dataset.

In general, the report is very cleanly written. It was easy to follow along. It is valuable that the authors include their misconceptions and how they fixed it. The report clearly states the scope of reproducability and follows it. In their scope, the authors attempted to reproduce the paper from scratch. Then, they discovered that the original paper falls short on the explanation of the exact architecture being used. The authors consult the the sources included with the original paper and contact the original paper authors. Overall, this shows a healthy and admirable approach to scientific investigation and collaboration.

The report examines the architectural choices made in the original paper. The report experiments with the proposed approach and compares with a more standard architecture (Vaswani et al., 2017). This part of the report reveals insightful results that the standard architecture yields similar or even better results.

The report extends the original paper with an extra experiment on a new dataset.

Suggested improvements or extensions:
- As far as I understood, the report does not conduct the hyper-parameter search. Nor it tries to determine the stability of the hyper-parameters. This investigation would greatly improve the report.
- With an exception for the case study mentioned above, the report doesn't provide any ablation studies.
- The figures are almost impossible to read in black and white. I encourage to modify the figures to make them more accessible to people with black and white printers and the color blind.

Overall, this is a well-structured, well-written report that analyses the original paper. Furthermore, the report extends the original paper with the results on an extra dataset. I recommend accept.

**Familiar With The Original Paper:**

I have read the original paper

**Reproducibility Summary:**

Report has summary

---

### Official Review · AnonReviewer1 · 2021-03-01
**Good submission and well documented**

**Rating:** 7
**Confidence:** 5

**Review:**

The authors of this report aimed at reproducing the method presented in the paper "Satellite Image Time Series Classification with Pixel-Set Encoders and Temporal Self-Attention" published at CVPR 2020 by Garnot et al.
The authors not only did they attempt to reproduce the code and evaluate it on the dataset used in the original paper, but they also went on to use another dataset to expand upon the evaluation process, including some changes to the way the test set was selected in the original dataset.
Given that the authors of the original paper have made their code available allowed for a direct comparison between the reproduced code and the original code. In fact, the authors of this report have had some issues on some aspects of the use of the original transformer implementation, but that was resolved in a communication with the authors of the original paper and of the Transformer one. Minor discrepancies and issues with the implementation were also resolved via checking the provided github code of the original paper, hence confirming the reproducibility of the original paper.
Finally, although the report is well written overall, I would have expected the conclusion to be a bit more elaborate on what was easy and what did not work as expected; but that is a very minor issue.

**Familiar With The Original Paper:**

I have read the original paper

**Reproducibility Summary:**

Report has summary

---

### Decision · Program_Chairs · 2021-03-31

**Decision:**

Accept

**Comment:**

Selected for ReScience-C Journal Publication.